# A Risk Factor Analysis of SARS-CoV-2 Infection in Animals in COVID-19-Affected Households

**DOI:** 10.3390/v15030731

**Published:** 2023-03-11

**Authors:** Evelyn Kuhlmeier, Tatjana Chan, Marina L. Meli, Barbara Willi, Aline Wolfensberger, Katja Reitt, Julia Hüttl, Sarah Jones, Grace Tyson, Margaret J. Hosie, Yury Zablotski, Regina Hofmann-Lehmann

**Affiliations:** 1Clinical Laboratory, Vetsuisse Faculty, Department of Clinical Diagnostics and Services, and Center for Clinical Studies, University of Zurich, Winterthurerstrasse 260, 8057 Zurich, Switzerland; tchan@vetclinics.uzh.ch (T.C.); mmeli@vetclinics.uzh.ch (M.L.M.); rhofmann@vetclinics.uzh.ch (R.H.-L.); 2Clinic for Small Animal Internal Medicine, Vetsuisse Faculty, University of Zurich, Winterthurerstrasse 260, 8057 Zurich, Switzerland; bwilli@vetclinics.uzh.ch; 3Department of Infectious Diseases and Hospital Epidemiology, University Hospital Zurich, Rämistrasse 100, 8091 Zurich, Switzerland; aline.wolfensberger@usz.ch; 4Center for Laboratory Medicine, Veterinary Diagnostic Services, Frohbergstrasse 3, 9001 St. Gallen, Switzerland; katja.reitt@zlmsg.ch (K.R.); huttljulia@gmail.com (J.H.); 5School of Veterinary Medicine, College of Medical, Veterinary and Life Sciences, University of Glasgow, Bearsden Road, Glasgow G61 1QH, UK; sarah.jones.4@glasgow.ac.uk (S.J.); g.tyson.1@research.gla.ac.uk (G.T.); 6MRC-University of Glasgow Centre for Virus, College of Medical, Veterinary and Life Sciences, University of Glasgow, Bearsden Road, Glasgow G61 1QH, UK; 7Clinic of Small Animal Medicine, Centre for Clinical Veterinary Medicine, LMU Munich, 80539 Munich, Germany; y.zablotski@med.vetmed.uni-muenchen.de

**Keywords:** SARS-CoV-2, COVID-19 households, risk factor analysis, RT-qPCR, one health

## Abstract

A higher prevalence of SARS-CoV-2 infections in animals that have close contact with SARS-CoV-2-positive humans (“COVID-19 households”) has been demonstrated in several countries. This prospective study aimed to determine the SARS-CoV-2 prevalence in animals from Swiss COVID-19 households and to assess the potential risk factors for infection. The study included 226 companion animals (172 cats, 76.1%; 49 dogs, 21.7%; and 5 other animals, 2.2%) from 122 COVID-19 households with 336 human household members (including 230 SARS-CoV-2-positive people). The animals were tested for viral RNA using an RT-qPCR and/or serologically for antibodies and neutralizing activity. Additionally, surface samples from animal fur and beds underwent an RT-qPCR. A questionnaire about hygiene, animal hygiene, and contact intensity was completed by the household members. A total of 49 of the 226 animals (21.7%) from 31 of the 122 households (25.4%) tested positive/questionably positive for SARS-CoV-2, including 37 of the 172 cats (21.5%) and 12 of the 49 dogs (24.5%). The surface samples tested positive significantly more often in households with SARS-CoV-2-positive animals than in households with SARS-CoV-2-negative animals (*p* = 0.011). Significantly more animals tested positive in the multivariable analysis for households with minors. For cats, a shorter length of outdoor access and a higher frequency of removing droppings from litterboxes were factors that were significantly associated with higher infection rates. The study emphasizes that the behavior of owners and the living conditions of animals can influence the likelihood of a SARS-CoV-2 infection in companion animals. Therefore, it is crucial to monitor the infection transmission and dynamics in animals, as well as to identify the possible risk factors for animals in infected households.

## 1. Introduction

A higher prevalence of SARS-CoV-2 infections in companion animals from households with SARS-CoV-2-infected household members compared with companion animals without known virus exposure has been demonstrated in different countries. A study from France found a seroprevalence of 21.3% in positive animals from SARS-CoV-2-positive households; the difference between dogs (15.4% positive) and cats (23.5% positive) was not significant. These results showed an eight-fold higher risk of infection for the animals from affected households compared with the animals without SARS-CoV-2 exposure [1]. Similar results were also found in studies in the USA, where 17.6% of cats, 1.7% of dogs [2], and, overall, 17% of companion animals [3] in SARS-CoV-2-affected households were infected.

What factors are decisive in determining whether animals living in households where humans test positive become infected? In humans, several factors and behaviors have been identified that increase their risk of infection with SARS-CoV-2. In a Brazilian study, age, mixed race, high school education, a low income, and contact with infected people were associated with a higher risk of a SARS-CoV-2 infection [4]. In another study, the male sex, adulthood, especially between 40 and 64 years and >70 years, black ethnicity, living in urban areas, chronic kidney diseases, and obesity were identified as risk factors for a positive SARS-CoV-2 test [5]. Besides demographic and socioeconomic factors [6], social factors also play a role. Social gatherings and social behaviors have different influences: visiting the gymnasium, journeys on a bus or a plane, and visiting the hairdresser were activities with an especially increased risk of infection in comparison with other social activities during the COVID-19 pandemic, as well as cultural social gatherings at religious spaces, sports stadiums, music concerts, movies, theaters, or amusement parks, which have a greater impact on the risk than small private meetings [7]. The risk factors for within-household human-to-human transmission include adult rather than child contacts, spousal contacts, contacts with comorbidities, symptomatic index cases, and households with only one in-contact individual [8].

Important measures to reduce the infection rate included shutdowns and public health interventions [9,10]. As an epidemiological measure, wearing surgical masks reduced the probability of SARS-CoV-2 transmission [11]. The possibility of indirect transmission has also been demonstrated [12]. SARS-CoV-2 can survive on the skin for up to nine hours [13], from where it may be transferred to the eye, mouth, or nose [14]. The presence of the virus on the hands and faces of infected people may also be relevant for its potential transmission to dogs and cats when they are petted by their owners. Furthermore, it is possible that infection may occur via contaminated surfaces upon which the virus is, depending on the material, stable for much longer than on the skin [13,15]. The presence of viral RNA on surfaces in SARS-CoV-2-affected households has been demonstrated [16], in some cases up to a month after the infected individuals ceased having clinical symptoms [17].

In a One Health context, it is important to understand how animals become infected with SARS-CoV-2 and what actions can be undertaken to reduce their risk of infection. Studies with small sample sizes from Canada and the USA on SARS-CoV-2 in animals found that the contact time between the owner and the animal and the proportion of positive people in the household were factors associated with a higher infection risk in animals; moreover, a reduction in the contact time between the infected humans and the animals decreased the risk of infecting the animal [3]. Furthermore, sleeping in an owner’s bed was identified as a risk factor for cats; in that study, animals with new owner-reported clinical signs tested positive significantly more often than clinically healthy animals [18]. A French SARS-CoV-2 risk factor analysis for companion animals revealed that the risk of infection for animals tended to be greater, although not significantly, in households with COVID-19 episodes [19]. Thus, only limited information concerning the risk factors for SARS-CoV-2 infection in companion animals is available, and there are no studies in Europe with significant results.

The current study aims to increase the knowledge about the potential transmission routes for SARS-CoV-2 to animals in infected households. We also aimed to determine the prevalence of the infection in animals and to identify the risk factors for infection. This information will form the basis upon which to provide pet owners with evidence-based options to reduce the risk of infection for their companion animals. Samples were collected from people, their companion animals, and the environment of the COVID-19 households. The animals’ SARS-CoV-2 infection status was determined using a RT-qPCR (real-time reverse transcriptase–polymerase chain reaction) and serology. Additionally, a questionnaire was used to assess the different types of human–animal contact and interaction, the differences between the household members (the age of the people and the number of SARS-CoV-2 affected household members, etc.), the general hygiene and animal hygiene standards, and the changes in these hygiene standards due to the SARS-CoV-2 infection in the household. Finally, these factors were analyzed using univariable and multivariable statistical analyses to evaluate the potential risk factors.

## 2. Materials and Methods

### 2.1. Recruiting SARS-CoV-2-Positive Households and Sampling

The recruiting of SARS-CoV-2-positive households with companion animals was performed by advertising the study with flyers and posters displayed at the University Hospital Zurich and the University of Zurich, as well as at the Animal Hospital Zurich. Further recruitment took place via the Center for Laboratory Medicine, Veterinary Diagnostic Services, St. Gallen, and the contact tracing program of the Canton of Zurich, as well as the homepages of the Veterinary Clinical Laboratory and the Canton of Zurich, with the support of cantonal medical and veterinary physicians. The inclusion criteria for participation were as follows: at least one confirmed SARS-CoV-2 infection in a person living in the household (“COVID-19-affected household”) and at least one companion animal living in the household. The intended minimum number of animals to be included in the entire COVID-19 household study was 300. The interested households then received further information on the study process. Upon their agreement to participate in the study, a consent form was requested from all the people in the household. Subsequently, a test kit with protective equipment was provided, which included disposable nitrile gloves (Lab Logistics Group GmbH, Meckenheim, Germany) and medical face masks (Zhejiang Longde Pharmaceuticals Co., Ltd., Hangzhou, China). The test material consisted of prelabelled 1.5 mL tubes (Sarstedt AG & Co. KG, Nümbrecht, Germany) filled with 300 µL of DNA/RNA Shield solution (Zymo Research Europe GmbH, Freiburg, Germany), which is known to inactivate SARS-CoV-2, cotton swabs (Heinz Herenz Medizinalbedarf GmbH, Hamburg, Germany), and cytobrushes (Deltalab S.L., Barcelona, Spain). The participants were also provided with detailed instructions and return envelopes. The materials were sent for three sampling timepoints (initial, follow-up 1, and follow-up 2) for each animal in the household and for five localizations (oral, nasal, fecal, fur, and animal bedding), as well as for the voluntary sampling of the persons in the household (oral and nasal). The sampling materials were packed separately in plastic bags (Minigrip^®^ Redline, Alpharetta, GA, USA) to avoid contamination. The collected samples were shipped in threefold packaging, in order to ensure the safety of all the people handling the package, at ambient temperature to the Clinical Laboratory, Vetsuisse Faculty, Zurich, for analysis. If the animals tested positive, more frequent sampling was planned, and more material was sent to the household for further follow-up sampling. Permission for blood collection (serum) was requested from the owner for the RT-qPCR-positive animals. This was performed after the animal tested RT-qPCR-negative. The complete process of the study is summarized in Figure 1.

### 2.2. Sample Analysis—Nucleic Acid Extraction, Molecular Analysis, and Confirmation

The screw lid tubes containing the swabs and the DNA/RNA Shield solution, which were packaged in plastic bags and a padded envelope, were unpacked in a laminar flow cabinet. As previously described [20,21,22], the screw lid tubes were rinsed with ethanol (70%), wiped, and then placed in an incubator (42 °C) and shaken for 10 min at 600 rpm. Subsequently, the tubes were centrifuged, the swabs inverted, the tubes were centrifuged a second time, and after that the swabs were removed.

The extraction of the total nucleic acid/ribonucleic acid (TNA/RNA) was performed according to the manufacturer’s instructions with either of the following: the MagNA Pure 96 instrument (Roche Diagnostics AG, Rotkreuz, Switzerland) with a MagNA Pure 96 DNA and Viral NA Small Volume Kit (Roche Diagnostics AG), used via the Viral NA Plasma ext lys SV protocol, or a MagNA Pure LC 2.0 instrument with a MagNA Pure LC Total Nucleic Acid High-Performance Kit (Roche Diagnostics AG, Rotkreuz, Switzerland Germany), or the QIAamp Viral RNA Mini Kit (Qiagen, Hilden Germany).

An ABI PRISM 7500 Fast Sequence Detection System (Applied Biosystems Foster City, CA, USA) was used for the SARS-CoV-2 molecular analysis. In total, two assays were run: firstly for the envelope gene (E-assay) and secondly for the RNA-dependent RNA polymerase sequence (RdRp-assay), if the first assay was positive or questionably positive. A RT-qPCR assay was performed as previously described [20].

The RT-qPCR result was considered positive if the threshold cycle (Ct) value in both of the assays was ≤38, questionably positive if the Ct values were >38 and <45, and negative if the Ct value was = 45.

The extracted TNA of the positive RT-qPCR test samples was sent to a veterinary reference laboratory in Switzerland, the Swiss Federal Institute of Virology and Immunology (IVI; Mittelhäusern, Switzerland), to confirm the results.

The owners were also asked to send in their own throat and nose swabs. Within the present study, these samples were collected to find out whether the animals from the owners with a higher virus load were more frequently infected than the animals living with the owners with lower loads. The owner samples were taken on the day the animal(s) within the same household were first sampled, or the day before/after this day. However, the human samples were only included in this study if they were taken within 10 days of the initial detection of the SARS-CoV-2 infection in the person, or if they tested positive on the day the animal(s) were first sampled. The viral loads were calculated using an in vitro-transcribed RNA control that contained the three concatenated sequences of RdRp, E, and nucleocapsid (N) SARS-CoV-2 genes (RNA_Wuhan_RdRp-E-N, provided by the IVI). The input viral RNA copy number of the samples was calculated using the efficiency, which was determined by the RT-qPCR amplification of a 10-fold serial dilution of the synthetic template. To evaluate the viral RNA load of the person, the mean copy number value was calculated from the result of the E-assay and the RdRp-assay (for details, see below).

### 2.3. Sample Analysis—Serological SARS-CoV-2 Testing

An in-house-developed enzyme-linked immunosorbent assay (ELISA) was used to detect the antibodies binding to the receptor-binding domain (RBD) of the SARS-CoV-2 spike protein. The test was performed as previously described [23].

The samples were also tested with a SARS-CoV-2 Surrogate Virus Neutralization Test Kit (sVNT; GenScript Inc., Piscataway, NJ, USA) for the neutralizing activity against SARS-CoV-2. The performance and cutoffs of the test have been described previously [23,24] and were performed according to the manufacturer’s protocol. The third test performed was a pseudotype-based virus neutralization assay (PVNT), which measures the neutralizing antibody activity. The HIV (SARS-CoV-2) pseudotypes bore the spike protein of one of four SARS-CoV-2 variants (Wuhan, Alpha, Delta, or Omicron), and the assay was performed as previously described [25]. The titers of the four assays were compared: the variants with no measurable titers were judged as negative, and for animals with at least one positive titer, the highest titer was assumed to correspond to the variant with which the animal had been infected.

### 2.4. Data Acquisition via Online Questionnaire

Each participating household was asked to complete a questionnaire. The questionnaire was made available to the households online via the LimeSurvey platform (LimeSurvey: An Open-Source survey tool, LimeSurvey GmbH, Hamburg, Germany; URL http://www.limesurvey.org (accessed on 6 February 2023)). The corresponding link was sent to all the households and, if desired, the questionnaire could also be filled out in paper form.

The questionnaire consisted of five different parts: (1) Questions characterizing the household, including the study number, date, number of animals in the household, the number, age and gender of the people in the household, which person(s) in the household takes care of the animal(s), which people were at risk and which people tested positive for SARS-CoV-2 and on what date, and if the household was familiar with the recommendations from the Swiss Federal Food Safety and Veterinary Office (FSVO) for SARS-CoV-2-infected animal owners. (2) The general hygiene behavior of the household members: how often the owners washed their hands, used a handkerchief when coughing and sneezing, and washed their hands after coughing and sneezing, and whether these behaviors changed after the onset of the SARS-CoV-2 infection. (3) Questions concerning the hygiene in regard to handling the animal(s): how often the toys, bed, and food bowl were washed, and what was used to clean the food bowls. (4) The household members’ contact times with the animals and whether/how the care of the animals had changed following the SARS-CoV-2 infection in the household. (5) Information about each animal in the household, with questions varying according to their species; these concerned the sex, age, living arrangements, previous illnesses, clinical signs, veterinary visits, and types of contact that were fostered with the respective animal, and how often these had taken place. The latter included close contact (having faces and hands licked, giving kisses, and giving treats), less close contact (sleeping in the same bed, lying together on the sofa, cuddling, walking alongside, playing, and staying in the same room), and indirect contact (feeding, removing droppings, and cleaning the dropping box). The complete questionnaire is available in the Appendix A. Based on data analyses of the questionnaire, all the animals were divided into two groups: a positive cohort that included all the serologically and/or RT-qPCR-positive and questionably positive-tested animals, and a negative cohort with the negative-tested animals.

### 2.5. Data Evaluation and Statistics

The variables of the answers from the questionnaire were compiled and analyzed using Excel (Microsoft). A recategorization for statistical purposes was carried out; the details can be found in Appendix B.

To evaluate the risk factors, first, a univariable logistic regression was performed for all the variables; second, all the variables with *p*-values < 0.20 in the univariable model were investigated further using multivariable logistic regression, as previously described [26]. The statistical significance was defined for the variables with *p*-values below 0.05. To indicate the strength of the independent variable’s association with the dependent variable, odds ratios were calculated. Due to the novel approach of our study, the *p*-values were not adjusted for multiple comparison to reduce the probability of a Type II error. All the statistical analyses were carried out in IBM SPSS Statistics (IBM Corp. Released 2021. IBM SPSS Statistics for Windows, Version 28.0. Armonk, NY, USA: IBM Corp).

## 3. Results

### 3.1. Participating COVID-19-Affected Households

A total of 122 COVID-19-affected households met the criteria, participated in the study from January 2021 to May 2022, filled out the questionnaire, and sent in swabs and/or serum samples from their animals for examination. The households that did not complete the questionnaire or did not submit samples were excluded. Individual animals in the households from which no samples were sent were also excluded, since it was not possible to determine whether they were SARS-CoV-2-negative or -positive. If a household completed the questionnaire more than once or started to fill it in more than once, the last available version was evaluated. Some of the households (*n* = 7) harboring the SARS-CoV-2 Delta variant were included in a recent publication detailing the molecular and serological analyses, without an evaluation of the data from the questionnaire [23].

Of the 336 people living in the 122 households that were analyzed, 230 tested positive for SARS-CoV-2. A total of 226 companion animals lived in these households, including 172 cats, 49 dogs, and 5 animals of other species: 2 horses, 2 rabbits, and 1 hamster.

### 3.2. Detection of SARS-CoV-2 Viral RNA and Confirmation of SARS-CoV-2 Infection via Serology

In total, 49 (37 cats, 12 dogs, and no animals of other species) of the 226 animals (21.7%) from 31 of the 122 households (25.4%) tested SARS-CoV-2-positive via an RT-qPCR of their oral, nasal, or fecal swabs and/or serologically.

The 49 positive animals included 37 of the 172 cats in the study (21.5%). Most of the cats tested positive via an RT-qPCR (*n* = 35). In six of them, the infection was confirmed by serology, and two additional cats tested positive only by serology (for detailed results, see Table 1). A total of 5 of the 37 positive cats (13.5%) were reported by their owners to have shown clinical signs potentially associated with SARS-CoV-2 infections (Table 1). These included respiratory and/or gastrointestinal signs in five animals.

The 49 positive animals also included 12 of the 49 dogs in this study (24.5%). They all tested SARS-CoV-2-positive using an RT-qPCR, and in two of the positive dogs, the infection was confirmed serologically (for detailed results, see Table 2). In total, four of the dogs (33.3%) showed clinical signs, as reported by their owners, including exhaustion (two dogs), respiratory signs (one dog), and gastrointestinal signs (one dog). All the RT-qPCR results of the positive cohort were confirmed by the Swiss reference laboratory.

### 3.3. Surface Samples

The swabs from the fur of 216 (168 cats and 48 dogs) of the 226 animals that were included in the study were available for SARS-CoV-2 testing (Figure 2).

Of the 133 SARS-CoV-2-negative cats (oral/nasal/fecal samples) with fur samples taken, 53 (39.8%) had positive fur samples. In the population of SARS-CoV-2-positive cats, this value was significantly higher, at 34/35 cats (97.1%) (*p* < 0.0001).

Among the dogs, 8 of the 36 with negative animal samples (oral/nasal/fecal samples) had positive fur samples (22.2%). Of the 12 SARS-CoV-2-positive dogs with fur samples taken, all 12 (100%) had positive fur samples. Thus, the fur swabs were also significantly more often positive in the dogs if the animal had also tested positive (*p* < 0.0001).

Bed swabs were provided from 214 of the animals (166 cats and 48 dogs) (Figure 3). In the cats with bed samples taken, 67.2% (88/131) of the SARS-CoV-2-negative cats and 97.1% (34/35 cats) of the SARS-CoV-2-positive cats showed a positive result for SARS-CoV-2 in their respective bed swabs. In the dogs with bed samples taken, 52.8% (19/36 dogs) of the negative dogs and 100% (12/12) of the positive dogs showed a positive result for SARS-CoV-2 in their respective bed samples. For both cats (*p* < 0.0004) and dogs (*p* = 0.0037), the animal beds were more frequently positive for the SARS-CoV-2-positive animals compared with the negative animals.

In summary, at the household level, the positivity rate of the surface samples in the current study was significantly higher in the households with SARS-CoV-2-positive animals (29/30; 96.7%—1 not available) than in the households with negative animals (59/89; 66.3%—2 not available; *p* < 0.001). If the surface samples (the fur and the bed of the animal) were positive, the animals were more frequently infected than when no positive surface samples were detected in the household.

### 3.4. Statistical Evaluation of the Questionnaire

#### 3.4.1. Univariable Logistic Regression

First, a univariable analysis of the variables that were under investigation was carried out. All the variables with a *p*-value ≤ 0.2 can be found in Table 3. At the household level, the influence of the number of positive people and the influence of the presence of children in the household was investigated. For the animals living with more than one SARS-CoV-2-positive person in the household, their risk of infection was significantly higher (OR 2.5; 95% CI 1.0–6.0) than that for the animals in SARS-CoV-2 households with only one positive person (*p* = 0.041, Table 3). The presence of minors in the households was significantly associated with an increased risk of pets being infected (OR 3.0; 95% CI 1.3–6.8; *p* = 0.011), compared with the households without minors. The frequency of the washing of the animals’ equipment (toys, beds, and bowls) had no significant influence on the risk of infection for the companion animals (*p* = 0.428; 0.490; and 0.801).

The contact time of the people with the animals in the household provided a significant result (*p* = 0.043), with a contact time of 10 min to 2 h per day having the highest risk of infection for the animals (36% positive; OR 2.0; 95% CI 1.2–3.2), compared with shorter or longer contact times (22% positive). The frequency with which hands were washed with soap per day had no significant effect on the infection risk of the animals (*p* = 0.113), and there were no associations identified for the frequency of washing hands after coughing (*p* = 0.741) or sneezing *(p* = 0.638).

The animal-specific parameters were analyzed separately for cats and dogs (Table 3):

Cats: The cats that had outdoor access for at least two hours per day were at a significantly lower risk of infection (3.5% positive; OR 0.4; 95% CI 0.099–1.58) than the cats with less available outdoor access (18% positive; *p* = 0.015). In addition, the different types of direct and indirect contact were tested for significance. If the animals licked the hands (*p* = 0.046) or the faces of their owners often or very often, they had a higher risk of infection (31% and 61% positive, respectively) than if they showed these behaviors rarely or not at all (both 17% positive; *p* = 0.014 and *p* ≤ 0.001, respectively; OR hands 2.6; 95% CI 1.2–5.4; OR face 7.6; 95% CI 2.7–21.5). The frequency of receiving treats was also significantly associated with the infection risk (*p* = 0.019); however, the cats that never or rarely received treats were more frequently positive (30%) than the cats that received treats often or very often (14%). The indirect contact parameters were also tested for significance. The cats that received food more than twice a day had a higher risk of infection (31% positive; OR 2.4; 95% CI 1.1–5.2) compared with the cats that received food less frequently (15%; *p* < 0.023). Moreover, the frequency at which the owners removed the cat droppings from their litterboxes or gardens was significantly associated with the infection risk of the cats (*p* < 0.001), with removal more than twice daily having a higher risk (63% positive; OR 9.8; 95% CI 3.8–25.2) compared with less frequent removal (16%). No *p*-values < 0.2 were found for the age of the cat (*p* = 0.578), the presence of pre-existing conditions (*p* = 0.883), owners giving kisses to the cat (*p* = 0.821), sleeping in the same bed (*p* = 0.203) or on the same sofa (*p* = 0.809) as the owner, cuddling (*p* = 0.676), playing with the cat (*p* = 0.231), or staying together in the same room (*p* = 0.612), so these parameters were not included in the multivariable analysis.

Dogs: In the univariable analysis, three of the variables achieved a *p*-value below 0.2 for the dogs: the presence of pre-existing conditions (0.066), owners giving kisses to the dogs (*p* = 0.112), and the frequency of feeding (*p* = 0.117). No *p*-values < 0.2 could be determined for age (*p* = 0.525), the frequency of licking the hands (*p* = 0.561) or faces of the owners (*p* = 0.999), the frequency of giving treats (*p* = 0.999), the frequency of sleeping in/on the same bed (*p* = 0.675) or sofa (*p* = 0.813), cuddling (*p* = 0.343), playing (*p* = 0.838), staying in the same room (*p* = 0.978), and cleaning up droppings (*p* = 0.684), so these were not included in the multivariable analysis.

#### 3.4.2. Multivariable Binary Logistic Regression

For all the variables with a *p*-value of <0.2, a multivariable logistic regression was performed to further define their significance (Table 4). If minors lived in the household, the risk of infection for the animals was significantly higher compared with the households without minors (*p* = 0.018).

Further significance was found only for cats, and not for dogs. The cats that stayed exclusively indoors or had limited access to the outside for less than two hours per day, or only on a balcony or terrace, were significantly more likely to contract SARS-CoV-2 than the cats with more than two hours of outdoor access per day (*p* = 0.045). Significance was also found for indirect contact: the cats from which the droppings were removed more frequently than twice per day were significantly more likely to become infected than the cats from which the droppings were removed less frequently (*p* = 0.007).

### 3.5. Owner Samples

Owner samples were available for testing from 100 of the 336 people from 68 of the 122 participating COVID-19 households. This included 61 people from the households in which the companion animals tested negative, and 39 people from the households where the companion animals tested positive.

The viral RNA loads were significantly higher in the samples from the people living with positive companion animals than from the people with no positive animals in their household (median viral loads: 1.8 × 10^6^ copies/reaction versus 1.5 × 10^5^ copies/reaction; Mann–Whitney U test *p* = 0.0085) (Figure 4).

## 4. Discussion

In this prospective study, we investigated the risk factors for the infection of companion animals living within households with SARS-CoV-2-infected people (COVID-19 households). In Switzerland, as in many other countries, a high proportion of households (often more than 40 percent) keep companion animals, mostly cats and dogs, and pets are an important part of their owners’ lives. Since the beginning of the COVID-19 pandemic in 2020, the majority of all households’ inhabitants underwent one or more SARS-CoV-2 infection, and, in particular, in 2020 people had to isolate themselves at home, where they were in close contact with their pets. Very few studies, with only a limited number of participants, have investigated the human–animal interactions within COVID-19 households and the infection risk for these companion animals. Given the widely propagated and urgently requested One Health approach, studies investigating not just human SARS-CoV-2 infections, but companion animal infections and contamination in the environment, provide important epidemiological information. In our study, we assessed a total of 122 affected households with 336 persons and 226 companion animals, recruited in 2021 and 2022. Several factors associated with an increased risk of infection in cats and dogs were identified. These will be addressed below in more detail, including, but not limited to, the presence of minors in the affected households. For cats, access to the outdoors and, with that, the possibility of defecating in soil rather than using litterboxes indoors, significantly reduced the risk of a SARS-CoV-2 infection. A similar association is known with regard to the risk of feline coronavirus infections in cats [27,28], although the underlying mechanism differs somewhat between the transmission of these two diseases. Contrary to what we expected, some parameters were not confirmed as protective factors against animal infections, e.g., a high handwashing frequency.

One of the interesting risk factors identified and, to the best of our knowledge, not yet reported, is the presence of minors living in the household: companion animals tested positive significantly more often when minors under the age of 18 also lived in the COVID-19-affected household. One study reported that children could carry a high viral load without presenting clinical symptoms or with only mild clinical symptoms [29]; this makes the children a possibly unrecognized source of infection. Another study showed that children up to 14 years old were at a higher risk of being carriers of the virus than all other age groups [30]. Even studies in which a higher susceptibility of children to SARS-CoV-2 was not generally found have suggested that children have a special role in viral transmission, since they often have close and mixed contact with each other in kindergartens and schools [31]. In addition, children’s hygiene habits are often not as well established as those of adults, and children can only be obliged to quarantine to a limited extent for ethical reasons. Moreover, children also often have very close contact with companion animals [32], a fact that could further explain why the presence of minors in the household was a significant risk factor for SARS-CoV-2 transmission to pets.

In the univariable analysis, the number of SARS-CoV-2-positive people in the household was also directly associated with a higher infection risk for the animals. Pets tested positive significantly more often if they lived with more than one positive person, rather than in households with only one infected person. An increased risk of infection for the animals living with a higher number of infected people could be explained by a higher number of infection sources and virus particles in the household. A similar finding was reported by Goryoka et al. in 34 COVID-19-affected households in the USA [3]. These authors found that the infection risk for companion animals increased with the proportion of people with laboratory-confirmed COVID-19 infections in the household. In the current study, this factor of significance could not be confirmed in the multivariable analysis.

The persistence of SARS-CoV-2 has been reported and compared on various materials [15,33,34]. SARS-CoV-2 virus particles can remain viable in aerosols for up to three hours [33] and on some surfaces for longer: for 72 h on plastic and 48 h on stainless steel, and on others, such as copper (4 h) and cardboard (24 h), for shorter periods [33]. Infectious virus could be detected on mink fur for an unusually long time, up to 10 days, but for less than 1 day on cotton [35]. Ideally, persistence studies are conducted using virus isolation to determine infectivity. However, in the current study, this would have required the collecting of infectious material from the COVID-19-affected households, the rapid transport of the infectious material, and analyses in a BSL-3 facility. For logistical reasons, we were not able to assess infectivity, but instead determined the persistence of the viral RNA. Nonetheless, the positivity rate of the surface samples (the fur and animal bed samples) in the current study was significantly higher (*p* = 0.011) in the households with SARS-CoV-2-positive animals than in the households with SARS-CoV-2-negative animals. On the one hand, this might be because the positive surfaces were contaminated by the animals, i.e., an infected animal contaminating its environment by licking its fur and sleeping in its bed. An alternative hypothesis is that the household members contaminated the investigated surfaces. This can happen frequently if the household members have close contact with the animal (fur) and its environment, for example its bed. Whether surfaces play a role in the transmission of SARS-CoV-2 is not clearly understood: some studies support the idea that fomites are sources of indirect SARS-CoV-2 transmission [36,37]; on the other hand, there are arguments that surfaces are not of great importance to transmission in real-life conditions [38,39]. Neither of the two hypotheses have been proven, and both are possible. However, it is likely that there are more viral particles in the air, as well as on surfaces, if there is a high infection pressure in the household due to many infected people and/or animals. While a high number of viral particles might be a risk factor for the infection of animals and humans, the presence of viral RNA might instead be an indicator of the infection pressure in the relevant household.

For cats, several risk factors for SARS-CoV-2 infections were identified. The cats with no or only limited outdoor access (terrace/balcony or <2 h per day) had a significantly higher risk of infection than the cats with more abundant outdoor access. In indoor spaces, the virus can accumulate, leading to airborne transmission without direct contact between individuals [40]. Thus, the cats with ample access to the outdoors can potentially escape the high infection pressure more often and for longer time periods, which, in turn, can lead to less exposure for the cats to SARS-CoV-2 and a lower infection risk. The virus’ persistence in aerosols and on surfaces in the entire household might demonstrate that the time the cat spends in the same room as an infected owner is not necessarily decisive in whether the cat becomes infected (there was no significant association in the current study between these two parameters).

Next to airborne transmission, droplet transmission is a very effective way of spreading SARS-CoV-2. This takes place in close contact, especially from less than one meter away [41]. The distance is reduced when the animal licks the hands and faces of its owners or receives treats. Liquid particles from the mouth, nose, or eyes can be absorbed directly into the animal’s mouth, and this can lead to infection. Consequently, the cats that were allowed to lick their owners’ hands, and, particularly, their faces, had a higher risk of SARS-CoV-2 infection, as determined in the univariable analysis. While this makes perfect sense, the result could not be confirmed in the multivariable analysis, possibly due to the low numbers and the complexity of the situation in these households.

The cats in COVID-19 households that were fed more frequently than twice per day were significantly more likely to become infected with SARS-CoV-2 than the cats fed less frequently. Again, this was only observed in the univariable analysis. In contrast, if the cat owner removed the pet’s droppings more than twice a day from the litterbox or the garden, the cats were significantly more likely to be infected, as was also confirmed in the multivariable analysis. This latter observation, an association between the frequent cleaning of droppings and the positivity of the animals, could have different explanations. It should be considered that the infection of the animals could be the reason for the more frequent cleaning of the litterboxes, since SARS-CoV-2 infection in cats can cause gastrointestinal signs in the infected animals, including diarrhea, which could have resulted in the owners cleaning up after the animals more often. Alternatively, frequent feeding or cleaning up after the cat can be explained as risk factors by the fact that the infected owner comes into contact more often with surfaces with which the animal also has contact. We should also consider whether these surfaces play a different role in the transmission of infections for animals versus humans. Animals have more direct contact with surfaces through sniffing, licking, and touching with their nose and mouth, sometimes even directly with mucous membranes, or with paws that are then licked by the animal while cleaning itself. This makes infection through surfaces more likely for pets than humans. Alternatively, or additionally, it can be speculated that pet owners who take care of their cats more frequently also have more intensive contact with their animals in general. In dogs, one factor that tended to be associated with a higher infection risk (*p* = 0.066 in univariable analysis) was a pre-existing condition in the animal. Little is known so far about the pre-existing conditions that predispose cats and dogs to SARS-CoV-2 infection. However, some cases of animals with pre-existing conditions becoming ill with SARS-CoV-2 are known [42,43,44]. As dogs seem to be less susceptible to SARS-CoV-2 infection [45,46], pre-existing conditions might be more crucial here.

The dogs in the present study had pneumonia, renal insufficiency, borreliosis, and eosinophilic bronchopathy reported as pre-existing conditions. Further studies in more animals or meta-analyses of many studies will be necessary to confirm certain pre-existing conditions as risk factors for infection. The clinical signs in the dog population studied here were tiredness, diarrhea, and respiratory issues. These are common clinical signs of SARS-CoV-2 infections in humans [47], and have also been observed in animals infected with SARS-CoV-2 [48].

We also expected some direct forms of contact, such as owners giving kisses to their dogs or the frequency of feeding their dogs, to be significant risk factors. However, this could not be confirmed, possibly due to the relatively small dog population (49 dogs) included in the study.

For two variables, we obtained unexpected results: the frequency of giving treats to cats was higher for the cats that tested negative. It can be assumed that contact during the treat giving was not very close or was associated with different proximities. The sheer length of the contact time between the animal’s owner and the animal also does not seem to be decisive. The contact time between the owners and their animals was also significantly associated with infection risk in the univariable analysis, in that 10 min to 2 h of contact time had the highest risk of infection for the animal, compared with shorter and longer contact times. Besides the duration of the contact, it can be speculated that the type and intensity of the contact that occurred are also important. Furthermore, the viral load of the person infected with SARS-CoV-2 has to be taken into account; the viral load may differ, e.g., depending on whether the disease is mild or severe [49] and on the stage of the person’s infection. We analyzed the viral RNA loads of the pet owners according to the infection status of their pets. The owners of the SARS-CoV-2-positive companion animals had higher viral RNA loads than the owners of the negative pets. This observation is consistent with the results of a Brazilian study on dogs [50]. A high viral load from the owners could thus also be a risk factor for SARS-CoV-2 infection in pets. However, it must also be noted here that the Ct value changes dynamically throughout an infection [51]. Therefore, the time of the sample collection is very influential, and further investigations are required.

This study aimed to provide a basis upon which to make recommendations to SARS-CoV-2-positive pet owners on how to handle their pets in order to avoid SARS-CoV-2 transmission, but the possible actions for the affected owners must be formulated carefully. Giving the pet to another household without children or limiting the contact between the minors and the animals during the period of infection should be carefully considered. The cost-benefit factor of a possible infection, which is often without or with only mild clinical signs in animals, should be weighed against the welfare of the animal. However, it might be useful to teach all the household members, especially children, not to have such close contact with their companion animals, as they can also become infected.

The fur and pet bed samples from the positively tested animals were positive significantly more often than those from the negative animals, and because the frequency of the dropping removal was significantly associated with the risk of infection in the animals, this also shows that indirect contact and contaminated surfaces could have an impact on the risk of infection for animals, but further investigations to test this hypothesis are necessary.

As a prophylactic measure, it may be useful to allow animals, especially cats that are used to going outside, plenty of access to the outdoors, as this reduces the time spent in a potentially infectious environment. Additionally, confinement inside would cause stress to animals that are used to going outside.

Overall, human behavior and the living conditions of their companion animals influence the risk of their being infected with SARS-CoV-2. Behavioral recommendations are an important element in the context of the One Health complex.

## 5. Conclusions

In the current study, several factors that are significantly associated with an increased risk of infecting animals were identified using a multivariable analysis: minors in the household, the positivity of surface samples (fur and bed) and, for cats, no/limited outdoor access and a high frequency of dropping removal from the litterbox. These results can be used as a basis for providing recommendations for SARS-CoV-2 infected people that are in contact with companion animals.

## Figures and Tables

**Figure 1 viruses-15-00731-f001:**
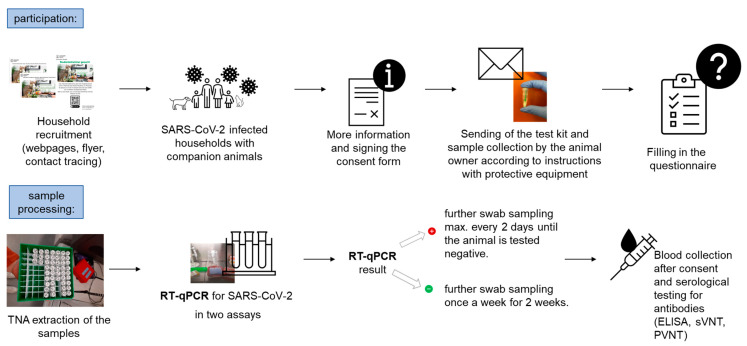
Study design for the recruitment process and sample analyses (TNA = total nucleic acid; ELISA = enzyme-linked immunosorbent assay; sVNT = surrogate virus neutralization test; and PVNT = pseudotype-based virus neutralization assay).

**Figure 2 viruses-15-00731-f002:**
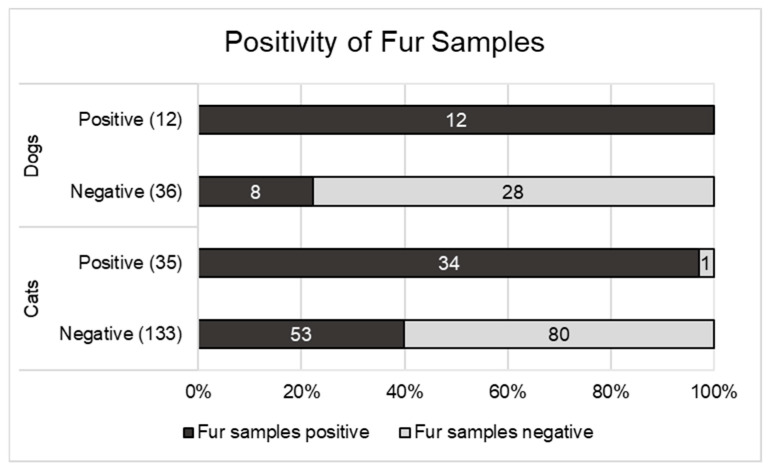
Distribution of positive fur samples for SARS-CoV-2-positive and SARS-CoV-2-negative dogs and cats.

**Figure 3 viruses-15-00731-f003:**
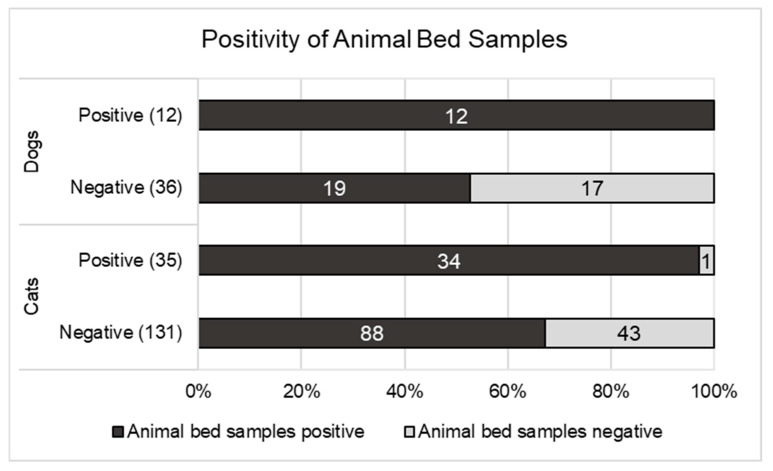
Distribution of positive bed samples for SARS-CoV-2-positive and SARS-CoV-2-negative dogs and cats.

**Figure 4 viruses-15-00731-f004:**
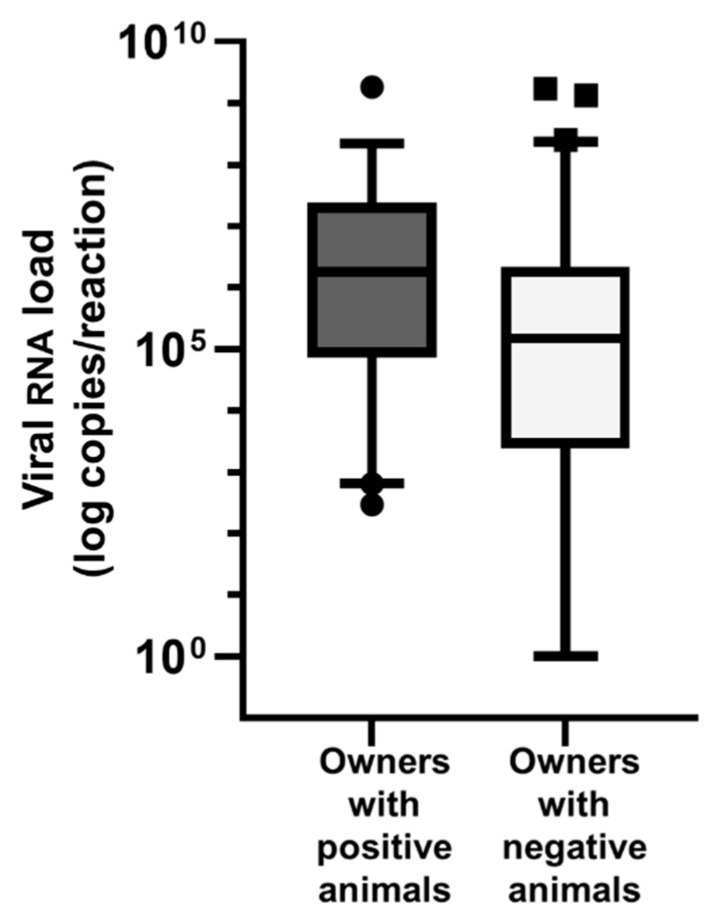
Relationship between the viral RNA load of samples from people and the positivity status of animals in the same household. A mean copy number value was calculated for each sample from the RT-qPCR results of the E-assay and the RdRp-assay. Box and whiskers plot: boxes extend from 25% to 75%, whiskers from 5% to 95%. Owners with SARS-CoV-2 positive animals had significantly higher viral RNA loads than owners with SARS-CoV-2 negative animals (*p* = 0.0085).

**Table 1 viruses-15-00731-t001:** Overview of the 37 SARS-CoV-2-positive cats: their ages, pre-existing conditions, clinical signs, RT-qPCR results from different samples, and serology results.

Animal ID	Age	Pre-Existing Conditions *	Clinical Signs *	RT-qPCR Results					Serology
				Oral	Nasal	Fecal	Fur	Bed	
Cat 22.2	N/A	Not known	Unsure	+	+	N/A	+	+	+
Cat 26.2	13	No		+	-	N/A	+	+	N/A
Cat 29.3	5	No		+	+	N/A	+	+	N/A
Cat 35.1	7	No	Respiratory sounds	-	+	-	+	+	N/A
Cat 36.1	19	Hyperthyroidism and kidney disease		+	+	+	+	+	+
Cat 38.2	N/A	*Giardia* (gastrointestinal)	Ocular discharge, vomiting, tiredness/listlessness, and reduced appetite	+	+	+	+	+	N/A
Cat 39.1 Cat 39.2	7 13	No No	Nasal discharge Unsure	+ +	+ +	- +	+ +	+ +	+ +
Cat 41.3	4	No		-	+	-	+	+	N/A
Cat 43.1 Cat 43.2 Cat 43.3 Cat 43.4 Cat 43.5 Cat 43.6 Cat 43.7 Cat 43.8 Cat 43.9	17 9 9 5 5 3 <1 <1 <1	No No Blindness No No No No No No		+ + - + + + + + -	+ + - + + + + + -	+ + + + + + + + +	+ + + + + + + + +	+ + + + + + + + +	N/A N/A N/A N/A N/A N/A N/A N/A N/A
Cat 47.2	6	No		-	+	-	+	+	N/A
Cat 49.1 Cat 49.2	11 17	Not known Not known		- +	- -	+ +	+ +	+ +	+ +
Cat 50.1	1	No		+	+	-	+	+	N/A
Cat 53.1	12	No		-	-	+	+	+	N/A
Cat 55.1	5	No		-	+	N/A	+	+	N/A
Cat 67.1	10	No	Vomiting	-	+	-	-	-	N/A
Cat 68.2	2	No		-	-	+	+	+	N/A
Cat 106.1 Cat 106.2	5 5	No No		- -	+ +	+ +	+ +	+ +	- -
Cat 140.1	10	No	Vomiting	-	+	+	+	+	N/A
Cat 152.1	5	No		-	+	N/A	+	+	-
Cat 154.2	2	No		-	-	+	+	+	N/A
Cat 169.2	<1	Eye infection		+	-	-	+	+	N/A
Cat 174.1 Cat 174.2	N/A N/A	No No		N/A N/A	N/A N/A	N/A N/A	N/A N/A	N/A N/A	+ +
Cat 179.1	5	No		+	-	-	+	+	N/A
Cat 159.1.1 Cat 159.1.5	12 2	Degenerative changes No		+ -	- +	+ +	+ +	+ +	N/A N/A
Total		5	5	19	24	22	34	34	8

N/A: not available; +: positive; -: negative; * as reported by animal owners.

**Table 2 viruses-15-00731-t002:** Overview of the 12 SARS-CoV-2-positive dogs: their ages, pre-existing conditions, clinical signs, RT-qPCR results from different samples, and serology results.

Animal ID	Age	Pre-Existing * Conditions	Clinical Signs *	RT-qPCR Results					Serology
				Oral	Nasal	Fecal	Fur	Bed	
Dog 22.1	N/A	Unknown	Unsure	+	+	-	+	+	+
Dog 29.1	<1	Pneumonia	Tiredness/listlessness	+	+	+	+	+	+
Dog 32.1	14	Renal insufficiency	Diarrhea (unsure)	-	+	-	+	+	N/A
Dog 37.1	9	Borreliosis	Tiredness/listlessness	+	+	-	+	+	N/A
Dog 71.1 Dog 71.2	5 <1	No No		+ -	- +	- +	+ +	+ +	N/A N/A
Dog 79.1	5	No		-	+	-	+	+	-
Dog 87.1	<1	No		-	-	+	+	+	N/A
Dog 127.1	4	No		-	+	-	+	+	-
Dog 156.1	11	Unknown	Respiratory sounds, ocular discharge	-	-	+	+	+	-
Dog 159.1.6 Dog 159.1.7	7 3	No Eosinophilic bronchopathy		- -	+ -	+ +	+ +	+ +	- -
Total		4	4	4	8	6	12	12	2

N/A: not available; +: positive; -: negative; and * as reported by animal owners.

**Table 3 viruses-15-00731-t003:** Variables with a *p*-value < 0.2 in the univariable binary logistic regression model.

Variables HOUSEHOLDS	Modalities	All Households (122 Households) ^1^	Households with Negative-Tested Animals (91)	Households with Positive-Tested Animals (31)	Univariable Regression
Number of positive household members	1 >1	55 67	46 (37.7%) 45 (36.9%)	9 (7.4%) 22 (18.0%)	0.041
Minors in the household	No Yes	75 47	62 (50.8%) 29 (23.8%)	13 (10.7%) 18 (14.7%)	0.011
Variables HOUSEHOLD- MEMBERS	Modalities	All household members (336 people)	Household members from households with negative-tested animals (241)	Household members from households with positive-tested animals (95)	Univariable regression
Contact time of the household members with the animals	<10 min a day 10 min–2 h a day 2–8 h a day >8 h a day	16 150 110 58	13 (3.9%) 96 (28.7%) 83 (24.8%) 47 (14.1%)	3 (0.9%) 54 (16.2%) 27 (8.1%) 11 (3.3%)	0.043
Handwashing with soap	≤4 times/day 5–6 times/day ≥7 times/day	94 104 125	65 (20.1%) 68 (21.1%) 97 (30.0%)	29 (9.0%) 36 (11.1%) 28 (8.7%)	0.113
Variables CATS	Modalities	All cats (172 cats)	SARS-CoV-2-negative cats (135)	SARS-CoV-2-positive cats (37)	Univariable regression
Outdoor access	Exclusively in the flat Access to balcony/terrace or <2 h/day ≥2 h per day outside	19 90 63	15 (8.7%) 63 (36.6%) 57 (33.2%)	4 (2.3%) 27 (15.7%) 6 (3.5%)	0.015
Licking hands	Never/rarely Often (daily)/very often (several times a day)	116 54	97 (57.0%) 36 (21.2%)	19 (11.2%) 18 (10.6%)	0.015
Licking face	Never/rarely Often (daily)/very often (several times a day)	152 18	126 (74.1%) 7 (4.1%)	26 (15.3%) 11 (6.5%)	<0.001
Receiving treats	Never/rarely Often (daily)/very often (several times a day)	71 97	50 (29.8%) 83 (49.4%)	21 (12.5%) 14 (8.3%)	0.019
Receiving food *	1–2x daily >2x daily	98 62	83 (51.9%) 43 (26.9%)	15 (9.4%) 19 (11.8%)	0.023
Removing droppings of the cat (garden/litterbox)	≤2x daily >2x daily	144 24	123 (73.2%) 9 (5.4%)	21 (12.5%) 15 (8.9%)	<0.001
Cleaning litterbox (e.g., changing sand, washing, and disinfecting)	<1x per week 1x per week >1x per week	87 39 33	73 (45.9%) 27 (17.0%) 27 (17.0%)	14 (8.8%) 12 (7.5%) 6 (3.7%)	0.166
Variables DOGS	Modalities	All dogs (49 dogs)	SARS-CoV-2-negative dogs (37)	SARS-CoV-2-positive dogs (12)	Univariable regression
Pre-existing conditions	Yes No	8 34	4 (9.5%) 28 (66.7%)	4 (9.5%) 6 (14.3%)	0.066
Receiving kisses	Never/rarely Often (daily)/very often	31 18	21 (42.9%) 16 (32.6%)	10 (20.4%) 2 (4.1%)	0.112
Receiving food *	1-2x daily Over 2x daily	29 19	25 (52.1%) 12 (25.0%)	4 (8.3%) 7 (14.6%)	0.072

^1^ Households with no responses or the answer “I do not know” to the questions are not listed. * Responses of less than 1x feeding per day are excluded, as it can be assumed that the respondent misunderstood this question and only referred to him/herself.

**Table 4 viruses-15-00731-t004:** Risk factors for animals in COVID-19-affected households after multivariable logistic regression of variables with a *p*-value < 0.2 in the univariable logistic regression.

Variables	Modalities	Odds Ratio (OR)	*p*-Value	95% Confidence Interval for OR
**Household**				
Number of positive Household members	1 >1	- 2.326	- 0.065	Reference (0.948–5.707)
Minors in the household	No Yes	- 2.795	- **0.018**	Reference (1.192–6.552)
**Household members**				
Contact time of the household members with the animals	Under 10 min a day 10 min–2 h a day 2–8 h a day Over 8 h a day	- 2.521 1.523 1.088	- 0.165 0.537 0.908	Reference (0.682–9.313) (0.401–5.790) (0.261–4.535)
Handwashing with soap	≤4 times/day 5–6 times/day ≥7 times/day	- 1.227 0.684	- 0.509 0.227	- (0.669–2.248) (0.369–1.267)
**Cats**				
Outdoor access	Exclusively in the flat Access to balcony/terrace or outside <2 h/day ≥2 h	- 0.494 0.172	- 0.331 **0.045**	Reference (0.119–2.046) (0.031–0.964)
Licking hands	Never/rarely Often (daily)/very often (several times a day)	- 0.973	- 0.968	Reference (0.265–3.581)
Licking face	Never/rarely Often (daily)/very often (several times a day)	- 2.385	- 0.336	Reference (0.405–14.039)
Receiving treats	Never/rarely Often (daily)/very often (several times a day)	- 0.446	- 0.114	Reference (0.164–1.213)
Receiving food	1–2x daily Over 2x daily	- 1.582	- 0.413	Reference (0.527–4.748)
Removing droppings of the cat (garden/litterbox)	≤2x daily >2x daily	- 6.056	- **0.007**	Reference (1.639–22.373)
Cleaning litterbox (e.g., changing sand, washing, and disinfecting)	<1x per week 1x per week >1x per week	- 0.834 0.576	- 0.772 0.428	Reference (0.244–2.851) (0.147–2.257)
**Dogs**				
Pre-existing conditions	No Yes	- 3.488	- 0.212	Reference (0.491–24.776)
Receiving kisses	Never/rarely Often (daily)/very often	- 0.179	- 0.131	Reference (0.19–1.673)
Receiving food	1–2x daily Over 2x daily	- 3.378	- 0.153	Reference (0.637–17.920)

Significant values (*p* < 0.05) are marked in **bold type**; and significant variables are highlighted in yellow.

## Data Availability

For some households, more detailed case descriptions can be found in the publication “Kuhlmeier, E.; Chan, T.; Agüí, C.V.; Willi, B.; Wolfensberger, A.; Beisel, C.; Topolsky, I.; Beerenwinkel, N.; Stadler, T.; Swiss SARS-CoV-2 Sequencing Consortium; Jones, S.; Tyson, G.; Hosie, M.J.; Reitt, K.; Hüttl, J.; Meli, M.L.; Hofmann-Lehmann, R. Detection and Molecular Characterization of the SARS-CoV-2 Delta Variant and the Specific Immune Response in Companion Animals in Switzerland. *Viruses*
**2023**, *15*, 245” [23].

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
