# Peer review of "A Risk Factor Analysis of SARS-CoV-2 Infection in Animals in COVID-19-Affected Households"

_viruses, 2023, doi:10.3390/v15030731_

Round 1

Reviewer 1 Report

It is an interesting analysis of SARS CoV-2 ecology: dissemination in household between owner and animal, the time of viral infectivity on different surfaces etc.

Author Response

We thank the author for this comment. The manuscript had been checked by the paid editing service prior to submission. It was now additionally edited for English language by two native English-speaking colleagues (co-authors).

Reviewer 2 Report

The article 'Risk factor analysis of SARS-CoV-2 infection in animals in COVID-19-affected households' is well constructed, the introduction is balanced, data size is small and unbalanced but still justifies the cause of the study. I have a few minor comments,

the authors wrote about the posting of samples to the labs, please write about safety concerns for ordinary people like postmen in detail, secondly, should we discuss 5 other animals in a bit more detail or exclude them? I am ok if you leave it in its current state too.

Author Response

We thank the reviewer for this comment. We added a sentence about safety concerns for ordinary people. Furthermore, the DNA/RNA Shield solution (Zymo Research Europe GmbH, Freiburg, Germany) is known to inactivate SARS-CoV-2 virus, therefore no viable virus was shipped by mail. The five described other animals are important to show the data in total, because they also lived together in households with other companion animals and fulfilled all inclusion criteria.

Reviewer 3 Report

Thanks for the invitation. This study focuses on one of the ongoing hot topics, but some modifications need to be made. 

comments are:

what about sample size calculation 

please add the strength and limitations of this study

The manuscript is well written. But the objectives should be well clarified. 

 In my opinion, the section conclusion should be shortened and discussion section should be improved
